# Classical and Emerging Regulatory Mechanisms of Cytokinesis in Animal Cells

**DOI:** 10.3390/biology8030055

**Published:** 2019-07-26

**Authors:** Vikash Verma, Alex Mogilner, Thomas J. Maresca

**Affiliations:** 1Biology Department, University of Massachusetts, Amherst, MA 01003, USA; 2Courant Institute of Mathematical Sciences, New York University, New York, NY 10012, USA; 3Department of Biology, New York University, New York, NY 10012, USA; 4Molecular and Cellular Biology Graduate Program, University of Massachusetts, Amherst, MA 01003, USA

**Keywords:** cytokinesis, microtubules, ECT2, RhoA, centralspindlin complex, RhoA and actin waves, cortical excitability, aurora B kinase, polo kinase

## Abstract

The primary goal of cytokinesis is to produce two daughter cells, each having a full set of chromosomes. To achieve this, cells assemble a dynamic structure between segregated sister chromatids called the contractile ring, which is made up of filamentous actin, myosin-II, and other regulatory proteins. Constriction of the actomyosin ring generates a cleavage furrow that divides the cytoplasm to produce two daughter cells. Decades of research have identified key regulators and underlying molecular mechanisms; however, many fundamental questions remain unanswered and are still being actively investigated. This review summarizes the key findings, computational modeling, and recent advances in understanding of the molecular mechanisms that control the formation of the cleavage furrow and cytokinesis.

## 1. Introduction

Cytokinesis, the last step of cell division, physically divides the cytoplasm by forming an actomyosin contractile ring between the segregated chromosomes; thereby ensuring that each daughter cell receives a full set of chromosomes. Spatial and temporal control of cleavage furrow formation between daughter nuclei is crucial for preserving the ploidy of a cell [1]. Failure in cleavage furrow formation and cytokinesis can give rise to tetraploid cells, which can further lead to aneuploidy. Aneuploidy is believed to be a major feature of cancer development and chromosomal instability [2]. Therefore, a cell must ensure spatiotemporal positioning, maintenance, and completion of the cleavage furrow with high fidelity. In the following sections, we review different models that have been proposed to describe the mechanisms by which cells establish the cleavage plane, provide an overview of the key regulators that control spatiotemporal patterning of RhoA, discuss the utility of computational modeling, summarize our recent knowledge of propagating waves in cytokinesis and highlight some of the outstanding questions.

## 2. Classical Models of Cytokinesis

Classical micromanipulation experiments performed by Rappaport in sand dollar embryos (reviewed by [3]), demonstrated that the spindle apparatus specifies the position of the cleavage furrow. Supporting evidence for this conclusion came from an elegant experiment where physical displacement of the spindle apparatus caused regression of the old furrow and formation of a new furrow in relation to the new spindle position within minutes [4,5]; however, what part of the spindle signals to the cortex and how these signals are delivered to induce contractility is still a matter of debate. Nonetheless, it is widely accepted that the cleavage plane is positioned via microtubule(MT)-dependent mechanisms [6]. Three prominent models, namely 1) central spindle stimulation, 2) astral stimulation, and 3) polar relaxation have been proposed to contribute to defining the cleavage plane [7,8,9,10,11,12,13,14,15,16,17].

### 2.1. Central Spindle Stimulation Model

According to this model, signals emanating from the spindle midzone are responsible for cleavage plane specification [14] (Figure 1A). Multiple studies support the idea that the centralspindlin complex, which is composed of a Kinesin-6 family member, MKLP1 (also known as Pavarotti in *Drosophila* and ZEN-4 in *C. elegans*) and a Rho GTPase activating protein MgcRacGAP (known as RacGAP50C/Tumbleweed in *Drosohpila* and CYK-4 in *C.elegans*) accumulate at the central spindle during anaphase. Homodimers of MKLP1 and MgcRacGAP assemble to form a heterotetrameric complex, which generates a furrow-inducing downstream signal by recruiting Rho Guanine Nucleotide Exchange Factor (Rho-GEF) ECT2 (known as Pebble in *Drosophila* and ect-2 in *C. elegans*), to the central spindle (an overlapping MT zone that assembles between segregating chromosomes during anaphase, note the term ‘central spindle’ and ‘midzone’ refer to the same structure and have been used interchangeably in this article. Readers are advised not to confuse the term ‘midbody’ with ‘midzone’. Midbody is a relatively thin, sometime bow tie-shaped, and compact MT structure that forms late in the cytokinesis and marks the site of abscission).

MKLP1 was initially identified in dividing Chinese hamster ovary (CHO) cells [18], which gave rise to its original name—CHO1. Orthologs of MKLP1 were later identified in *Drosophila* [19] and *C. elegans* [20]. In accordance with its role in spindle midzone/central spindle formation, Pavarotti mutants displayed defects in microtubule bundling and caused cytokinesis failure because contractile rings failed to assemble in the mutant cells [19]. Similar phenotypes of ZEN-4 depletion were observed in *C. elegans* [20]. The binding partner MgcRacGAP (Male Germ Cell RacGAP) was identified in human testis and germ cell tumor extracts through a two hybrid screen [21]; the *C. elegans* ortholog, CYK-4 was later identified in a genetic screen [22]. Likewise, it was established that MgcRacGAP/CYK-4 is essential for central spindle assembly and cytokinesis [23,24] as the *cyk-4* temperature sensitive (ts) mutants (*t1689ts*) initiated furrow ingression but did not complete cytokinesis due to cleavage furrow regression; and mutations in the GAP domain of MgcRacGAP produced multinucleated cells, a hallmark of cytokinesis failure [23,24]. Since MKLP1 and MgcRacGAP together form a heterotetrameric complex, and their localization to the central spindle is dependent on each other, one would expect similar phenotypes for mutants of MKLP1 and MgcRacGAP; however, discrepancy in the severity of cytokinetic phenotypes as discussed above in the case of Pavarotti/ZEN-4 mutants and MgcRacGAP/CYK-4 mutants could stem from different mutational backgrounds that were used in the studies.

There is no doubt that the centralspindlin complex plays an essential role in cytokinesis and cleavage plane specification; however, these experiments do not provide conclusive evidence that the central spindle communicates with the cell cortex to specify the cleavage plane or plays a significant role in cytokinesis. Further insight into the direct role of the central spindle in cytokinesis came from multiple studies, which are summarized below. By exploiting micromanipulation techniques in a diving neuroblast cell of the grasshopper, Kawamura deformed the spindle into a U-shape, where the midzone and poles occupied opposite sides of the cortex. [25]. Kawamura observed increased cleavage stimulus near the midzone side, and argued that signaling cues for the cleavage furrow emanated from the midzone during anaphase and not from the asters. [26] showed that the central spindle can induce furrowing if the cortex and central spindle were in close proximity to each other. Another crucial piece of evidence was attained using cultured epithelial cells [14], where a physical barrier was introduced between the central spindle and the cortex to block the signals emanating from the central spindle. This manipulation resulted in production of binucleate cells and cytokinesis failure, leading to the proposal that the central spindle-mediated signals are required for cortical contractility and cleavage plane specification. Moreover, another study in mammalian cells showed that the continuous presence of midzone microtubules is required for successful cleavage [27]. It is believed that a diffusive positive signal originating from the central spindle specifies the cleavage plane [28]; however, the molecular nature of this signal remains elusive. Altogether, these studies supported the notion that the central spindle plays a vital role in transmitting furrow-promoting signals to the cell cortex. One major caveat for this model is that it fails to explain how a furrow-specifying signal(s) reliably reaches the cell cortex from the central spindle, which can be microns away from the plasma membrane.

In addition to the centralspindlin complex, another key regulator of cytokinesis is the chromosomal passenger complex (CPC), which is comprised of four proteins: survivin, borealin/dasra B, INCENP, and Aurora B kinase (ABK) [29,30]. CPC localizes to centromeres throughout mitosis prior to anaphase. At anaphase onset, with the help of MKLP2, a plus-end directed kinesin, the CPC re-localizes to the central spindle [31,32]. Midzone-localized CPC could be a potential source of a diffusive signal because ABK-mediated phosphorylation regulates many events during mitosis and cytokinesis. To investigate the spatial dynamics of ABK phosphorylation, Fuller et al. developed a FRET-based sensor in living cells [33]. FRET measurements and other supporting experiments suggested that CPC can produce an ABK-based phosphorylation gradient emanating from the midzone MTs during anaphase [33,34,35,36] which is now thought to provide a critical spatiotemporal cue for furrow positioning. The CPC component INCENP binds to actin and has also been shown to enrich CPC at the equatorial cortex to help with cleavage furrow ingression [32]. An accompanying proposal for this model is that ABK facilitates the formation of centralspindlin oligomers at the equatorial plasma membrane in *C. elegans* to initiate cleavage furrow formation [37]. In addition to CPC and the centralspindlin complex, many key regulatory proteins, such as Polo like kinase 1 (PLK1), Protein Regulator of Cytokinesis 1 (PRC1), and the Rho GTPase Guanine nucleotide-exchange factor (GEF) ECT2, accumulate at the central spindle (discussed later), which led to the proposal that the midzone microtubules provide a platform from which critical signaling cues for cleavage furrow specification originate. 

### 2.2. Astral Stimulation Model

This model posits that astral microtubules originating from mitotic asters induce furrow formation (Figure 1B). The famous “torus experiment” (Figure 1 D) performed by Rappaport provided the first evidence that positive furrow-specifying signals are delivered to the cortex along equatorial astral MTs [6,38]. To investigate what part of the spindle signals to the cortex, Rappaport used micromanipulation techniques to perforate a cell, which positioned the mitotic spindle to one side of the torus (Figure 1D). Surprisingly, the first furrow ingressed normally without any delay, but it produced a binucleate, horseshoe-shaped cell (Figure 1D). The binucleate, horseshoe-shaped cell, initiated a second round of furrow ingression, which proceeded normally and on time with the control cells. After the two spindles in the horseshoe-shaped cell entered anaphase two cleavage furrows formed between the segregating sister chromatids. This would be expected as these structures contained all the pertinent landmarks of a normal anaphase spindle: two asters, a midzone microtubule array, and segregating sister chromatids. Remarkably, however, a third furrow formed at the bend in the horseshoe between two asters from the separate spindles. This third cleavage furrow assembled between asters without a spindle midzone or chromosomes (Figure 1D), which led to the proposal that asters alone were sufficient to communicate with the cell cortex. Rappaport referred to this furrow as a “non-spindle furrow”, which is also famously known as ‘Rappaport cleavage’ (Figure 1D).

Based on the assessments that equatorial astral MTs are stabilized relative to polar MTs and that MKLP1 is a plus-end directed motor, Odell and Foe developed a computational model for furrow positioning positing that equatorial astral MTs become stabilized at anaphase onset [39] and, as a result, persist long enough to serve as tracks for the motor protein MKLP1 to deliver a positive furrowing signal to the equatorial cortex [40]. However, this assumption is not universal, and some reports indicate that contact between the cortex and the astral MTs is important for furrow initiation, and not the stabilization or dynamic state of MTs [8,41]. In addition, there is evidence that MKLP1 motor activity is dispensable for cleavage furrow initiation [42,43]. Thus, neither the stabilization of equatorial MTs nor MKLP1 motility are universally accepted features of the astral stimulation model. 

### 2.3. Polar Relaxation Model

In favor of this model, it has been proposed that signals emanating from the poles of the mitotic spindle induce relaxation of the polar cortex (Figure 1C). The underlying assumption for this model is that inhibitory signals originating from the poles and/or segregating chromosomes would be maximum at the poles and minimal at the equator. However, the molecular nature of this inhibitory signal is still under investigation. In mammalian cells, Murthy and Wadsworth showed that MT-derived inhibitory signals impede contractility near the polar cortex by blocking RhoA activity [44]. In *C. elegans*, the nature of this inhibitory signal has recently been shown to be dependent upon TPXL-1-based activation of Aurora A kinase [45], which mediates clearance of contractile ring components from the polar cortex. Another study in *Drosophila* and human cells found that inhibitory signals originate from a kinetochore-derived PP1-Sds22 phosphatase activity gradient during anaphase [46,47] that dephosphorylates and inactivates membrane stiffening proteins in the polar regions.

Although multiple mechanisms of cleavage plane specification have been proposed they are not mutually exclusive, and it has become evident that cells might have evolved redundant mechanisms to ensure the fidelity of cytokinesis as the process is central to the survival of cells.

## 3. Spatiotemporal Regulation of Cytokinetic Events

It is essential for cells to establish the cleavage furrow once the chromosomes are correctly segregated, failure in the timing could result in aneuploidy or polyploidy. To achieve this, cells have evolved a complex network of regulatory kinases. The best-known kinases that regulate important events in cytokinesis are: 1) Cyclin-dependent kinases (Cdks), 2) Polo-like kinases (Plks), and 3) Aurora kinases.

### 3.1. Cdks

High Cdk1 activity during metaphase phosphorylates key components of the cytokinetic apparatus such as MKLP1, MKLP2, PRC1, and INCENP to reportedly reduce their affinity for MTs [48,49,50,51]. However, during the metaphase-to-anaphase transition, Cdk1 activity drops as anaphase promoting complex (APC) inactivates Cdk1 through cyclin B degradation. As a result, inhibitory phosphorylation on MKLP2 is removed, allowing it to bind INCENP and transport CPC to the midzone. Concurrent loss of Cdk1 phosphorylation from PRC1 and MKLP1 allows them to bind, bundle, and stabilize MTs and localize to the midzone. Inhibition of Cdk1 activity with BMI-1026 results in precocious formation of a cleavage furrow, but key cytokinetic regulators MgcRacGAP, ECT2, and RhoA are required for precocious furrowing in HeLa cells [52]. Cdk1 activity has been suggested to inhibit myosin ATPase activity until the chromosomes have segregated through phosphorylation of its regulatory light chain [53,54]; however, in echinoderm embryos this might not be the case [55]. As the Cdk1 activity drops during anaphase, inhibitory phosphorylation on myosin regulatory light chain decreases and at the same time activating phosphorylation by myosin light-chain kinase on the myosin regulatory light chain increases many fold [56]. This spatial and temporal change in phosphorylation might trigger cytokinesis and contractility of actomyosin ring.

### 3.2. Polo-Like Kinase 1 (Plk1)

Plk1 is one of the best-studied kinases in mitosis and cytokinesis. Like the chromosomal passenger complex (CPC), Plk1 localizes to multiple locations. Substrate specificity is believed to be accomplished via its C-terminus polo box domain, which binds to phosphorylated (often by Cdk1 or Plk1 itself) sites on its substrates [57]. The gene for Plk1 was first identified in genetic screens in *Drosophila* and yeast [58,59] for mutants that were defective in cell division. Original identification of the polo gene in *Drosophila* suggested its role in the assembly of mitotic spindles as mutation in the polo gene led to the formation of monopolar spindles and abnormal centrosomes [58]. Use of polo kinase inhibitors at the metaphase-to-anaphase transition further reveled its role in cytokinetic furrow ingression [60,61]. Cells treated with Plk1 inhibitors were unable to accumulate RhoA at the equatorial cortex [60,61] due to the fact that ECT2 localization to the spindle midzone and MT plus-ends were abolished [60,61,62,63]. Apart from its role in cytokinesis, Plk1 activity has also been shown to be important for the timing of mitotic entry [64,65], kinetochore-MT attachment [64,66,67], centrosome maturation [68], and spindle elongation [57,60,63].

Plk1 localizes to the central spindle, which is mediated by Plk1-dependent phosphorylation of PRC1 [69] and requires Kinesin-4 (Klp3A) [62]. In addition to its localization to the central spindle, Polo kinase (Plk1 ortholog in *Drosophila*) has also been shown to localize to the MT plus-ends in a recent study [62]. Inhibition of Plk1 activity impedes its localization to the central spindle, but not to the MT plus-ends [60,61,62,63]. This indicates that factor(s) that recruit Polo kinase to the MT plus-ends is not mediated by PRC1 and kinesin-4. Future studies will be necessary to identify the factor(s) responsible for recruitment of polo kinase to the MT plus-ends and to determine if the MT plus-end localization pattern is conserved beyond *Drosophila*.

### 3.3. Aurora Kinases

Aurora kinases are a family of serine/threonine kinases that regulates many events in mitosis and cytokinesis [30,70,71]. The responsible genes (*IPL1* and *IPL2*) were identified in a genetic screen in budding yeast for mutants that were defective in chromosome segregation [72]; and in *Drosophila* the genes were identified in a screen that were defective in centrosome cycle [73].

Three major Aurora Kinases: Aurora A kinase (AAK), Aurora B kinase (ABK), and Aurora C kinase (ACK) have been implicated in cell division. These kinases are highly conserved and display similar structural features, consisting of an N-terminal domain, a protein kinase domain, and a C-terminal domain containing the degradation box (D-box) [74]. The N-terminal regulatory domain is more divergent and controls the distinct cellular localization patterns and, therefore, functions of each Aurora kinase [75]. Aurora A localizes to the centrosomes, spindle MTs, and midzone, and functions in centrosome maturation, separation and function [73,76,77], proper kinetochore-MT attachment [78,79,80,81,82], chromosome segregation [83], MT nucleation [84], and robust assembly of midzone MTs [85,86,87]. ABK functions as a part of the CPC [29]; and shows dynamic patterns of localization throughout mitosis. During metaphase, it is enriched at the inner centromere region between sister kinetochores. However, at the metaphase-to-anaphase transition, it dissociates from the centromeres and localizes to the midzone MTs, and eventually accumulates on the midbody region during telophase [31]. ABK also localizes to the equatorial cortex and MT plus-ends during cytokinesis [62,88].

Aurora Kinases regulate a wide range of activities during cell cycle progression; therefore, using genetic tools, to tease apart its specific function in the later stages of mitosis has proven difficult. Development of small molecule inhibitors against AAK and ABK made it easier to study its role in later stages such as cytokinesis. Inhibition of AAK in cells going through anaphase led to the conclusion that its activity is required for robust assembly of the midzone [81,85,86,87]. However, loss of AAK activity at the central spindle did not significantly change the dynamics or organization of myosin-II during cytokinesis in *Drosophila S2* cells [87]. Likewise, to examine post-metaphase functions of ABK in *Drosophila S2* cells; cells treated with Binucleine-2 (a specific inhibitor of ABK in *Drosophila*) resulted in production of large cells (an indication of cytokinesis failure) [89], and localization of *Drosophila* MKLP1 and Polo kinase was lost from the MT plus-tips within minutes [62], indicating ABK activity is required for cytokinesis [62,90,91]. Interestingly, depletion of Subito/*Drosophila* MKLP2 which localizes CPC to the midzone in various organisms [12,92,93] resulted in loss of phospho-ABK from the midzone; however, cells completed cytokinesis [62], suggesting that midzone based ABK activity is dispensable for cytokinesis in *Drosophila* S2 cells.

## 4. Rho GTPases in Cytokinesis

Key to the cytokinetic furrow formation is the localized activation of RhoA at the cortex (reviewed in [94]), which stimulates cortical contractility by assembling the actomyosin ring. Precocious activation of RhoA and furrow formation could result in aneuploidy or polyploidy; therefore, the activity of the RhoGEF ECT2, which activates RhoA by catalyzing the exchange of GDP for GTP is restrained in multiple ways to control the spatial patterning of RhoA activation. ECT2 is subject to auto-inhibition [95] via an intramolecular interaction between N-terminal BRCT domain and the C-terminus, which masks the catalytic C-terminal domain responsible for guanine nucleotide exchange [95]. Phosphorylation of the centralspindlin component MgcRacGAP by Plk1 recruits ECT2 to the midzone and activates it [60,61], which in turn activates RhoA resulting in formation of the cleavage furrow (Figure 1E). Cells expressing non-phosphorylatable mutants of MgcRacGAP disrupted ECT2 localization to the midzone and resulted in the failure of RhoA activation and cleavage furrow formation [61,96]. Likewise, depletion of ECT2 using RNAi in *C. elegans* disrupted embryonic cleavage formation [97] and mutations in *Drosophila* ECT2 (Pebble) caused cytokinesis failure owing to defects in contractile ring assembly [98]. Interestingly, artificial recruitment of ECT2 to the plasma membrane was sufficient for RhoA activation [99] and resulted in initiation of cytokinetic furrow even in interphase cells [100], indicating that the only requirement for RhoA activation is the cortical targeting of ECT2 [100].

The MgcRacGAP subunit of the centralspindlin complex contains a GTPase activating protein (GAP) domain that is conserved across species [23]. Although, the function of the GAP domain of MagRacGAP has been examined in various systems, there is some controversy regarding its GAP activity [101,102]. In vitro studies where GTP hydrolysis was used to measure the GAP activity indicated that MgcRacGAP/CYK-4 was a more efficient GAP for CDC42 and RAC1 than for RhoA [21,23,103]. Genetic studies provided further insights into the function of GAP activity; however, no consensus has emerged. Nonetheless, its cytokinetic functions can be summarized into three major categories: 

1) GAP activity is dispensable for cytokinesis: Goldstein et al. showed that in *Drosophila* neuronal cells the GAP activity of Tumbleweed (*Drosophila* MgcRacGAP) is dispensable for regulating cell division [104]. A GAP domain mutant (R417L), which is predicted to disrupt the GAP activity, was able to rescue the neuroblast proliferation phenotype in most of the cells; but was not fully functional in regulating axon growth. These results led to the conclusion that Tumbleweed’s GAP activity is required for axon growth regulation in post-mitotic neurons, but dispensable for cell division [104]. In a second study, Yamada et al, showed that in B lymphocytes, the GAP activity of MgcRacGAP is not required for cytokinesis [105]. 

2) GAP activity is required for RhoA activation: supporting evidence for this seemingly counter-intuitive possibility came from a study using *C. elegans* embryos expressing CYK-4 with a mutation (E448K) in its GAP domain, which caused defects in furrow ingression and myosin accumulation at the cleavage plane. The E448K mutant phenotypes was similar to that observed following partial RhoA depletion leading to the conclusion that cytokinetic defects in *cyk-4* mutant embryos stem from failure in RhoA activation [106]. Contrary to the study in neuronal cells, another study in *Drosophila* reported that the GAP activity in Tumbleweed/RacGAP50C is required for cytokinesis. To investigate the function of Tumbleweed, this study introduced a premature stop codon in the Tumbleweed gene. Cells expressing the truncated version of Tumbleweed failed to furrow, and key cytokinetic regulators, Pavarotti (*Drosophila* MKLP1), ABK, Pebble (*Drosophila* ECT2), and Anillin did not localize to the central spindle. To further isolate the specific function of the GAP domain of Tumbleweed, two kinds of deletions (∆ ^404^*EIE*^406^ or ∆ ^416^*YRL*^418^) were introduced within the GAP domain of Tumbleweed. *Drosophila* embryos expressing these deletions failed to rescue the cytokinesis failure phenotype. Binucleate cells were frequently observed in cells expressing either ∆ ^404^*EIE*^406^ or ∆ ^416^*YRL*^418^, leading to the conclusion that the GAP activity of Tumbleweed is required for cytokinesis [107]. However, it is not clear whether the cytokinesis defects observed in this study originated from RhoA activation failure. 

3) GAP activity negatively regulates RAC: supporting evidence for this came from temperature-sensitive, separation-of-function mutations in the GAP domain (E488K and T546I) of CYK-4 in *C. elegans* embryos, where mutant alleles caused cytokinesis defects that resembled the phenotype displayed by loss of the centralspindlin complex [108]. Remarkably, cytokinetic defects were rescued by depletion of the Rho GTPase, RAC, leading to the conclusion that the GAP domain of MgcRacGAP negatively regulate RAC [108]. In another study, a CYK-4 GAP mutant was exploited to understand the mechanisms of cell adhesion and cytokinesis. Results from this study supported the conclusion that CYK-4 GAP activity is required to block RAC1-dependent PAK1 and ARHGEF7 pathways [103].

The function of MgcRacGAP’s GAP domain is further complicated by findings from a study using HeLa cells where Minoshima et al. showed that ABK mediated phosphorylation of MgcRacGAP’s GAP domain at a key residue (S387) converts its specificity from RAC1 to RhoA [109]. However, the key serine residue is not conserved universally, and a phospho-mimetic mutant of S387 completely abolishes, rather than activates, the GAP activity of CYK-4 [101,103]. Discrepancies may be a consequence of various studies being performed in different organisms/mutational backgrounds and cell types where different functional redundancies may exist. Surprisingly, even in the same model organism, *Drosophila*, two different studies reached opposite conclusions—one claiming that the GAP activity is dispensable for successful cell division [104], and another arguing that the GAP activity is required for cytokinesis [107].

Nonetheless, one important point to note is that most of these studies were conducted in a mutational background that disrupted the GAP activity of MgcRacGAP/CYK-4/Tumbleweed, and authors did not extensively analyze the effects of these mutations on the stability of MgcRacGAP and its 3-D protein structure, which might affect the localization of MgcRacGAP as well as other regulatory proteins. These issues were recently addressed by [110], where authors show that *cyk-4* (or 749ts) mutants were defective in membrane binding. In addition, the authors provided multiple lines of evidence to show that the GAP domain of MgcRacGAP/CYK-4 contributes to RhoA-mediated contractility, presumably through ECT2 activation. In support of their conclusion, the authors showed direct interactions between ECT2 and the C-terminus of CYK-4 through a GST pull-down assay, although no affinity measurements and quantitation for this interaction were provided. Crucially, the authors did not provide any evidence that this interaction indeed activated the GEF activity of ECT2. Further investigation is clearly required to settle the biological relevance of the GAP activity of MgcRacGAP/CYK-4/Tumbleweed in cytokinesis.

Two other GTPases, namely RAC1 and CDC42, have been implicated in cytokinesis, both of which have been reported to inhibit formation of the cleavage furrow. RAC1 impedes cytokinesis by enhancing Arp2/3 complex-mediated assembly of branched actin networks [101]. Inhibition of RAC1 by the GAP activity of MgcRacGAP and concurrent activation of RhoA promotes cytokinesis by a formin-mediated pathway that supports elongation of unbranched actin filaments [111]. The role of CDC42 in cytokinesis remains elusive. In a recent study in budding yeast, it was shown that CDC42 activity oscillates during cell cycle progression [112], peaking during G1/S transition and anaphase and dropping during mitotic exit and cytokinesis [112]. Decreased CDC42 activity is required for localization of key regulatory proteins at the cleavage plane [112]. Addressing whether CDC42 plays similar roles in other organisms would be an interesting line of investigation.

## 5. Computational Modeling of Cytokinesis

Recently, Tom Pollard posed nine unanswered questions about cytokinesis [113]. All of those are being answered with various experiments, and some also benefitted from quantitative modeling. The questions that led to modeling are: What specifies the site of the furrow? How does the contractile ring assemble? How does the ring generate force? How does this force form the furrow? Here we discuss the utility of the modeling approach in understanding cytokinesis. The reader is also referred to two recent reviews [114,115].

### 5.1. What Specifies the Site of the Furrow?

Qualitative cartoons of the astral stimulation and polar relaxation models pose the question: given astral MT positions and lengths and cell shape, would greater signal be delivered by the astral MTs to the equatorial or polar regions of the cortex? To answer this, White and Borisy (1983) computed density of the astral MT plus ends at the cortex from two spindle asters in a spherical cell [116]. By assuming symmetric position of the spindle, stable MTs stretching all the way from the centrosome to the cortex and proportionality of the signal from an individual MT to an inverse power of the MT length, calculations predicted the maximal signal at the poles. However, the polar relaxation model predicted incorrectly the furrow positioning in a few experimental modifications of cellular shape [117]. Two studies [117,118] therefore proposed the modification of the White–Borisy model: there is a region in the middle of the spindle into which the MTs do not grow. Plus, signaling molecules do not “jump” from the cortex to the MT plus ends, but rather detach from the plus ends near the cortex, diffuse, and are spontaneously degraded in the cytoplasm. This modified model predicted global signal maxima at the equator, leading the authors to conclude that the astral stimulation model accounts for the experimental data. A later model [119] added the central spindle hypothesis and tested all three mitotic apparatus-dependent models in the unified framework, concluding that all three models are viable, and performance of respective mechanisms depends on many biochemical, geometric, and kinetic parameters.

A more recent detailed model of the furrow positioning [40] demonstrated computationally that a combination of three effects can explain a significant buildup of centralspindlin at the equator: 1) an indirect cortical relaxation due to centralspindlin binding to the sides of dynamically unstable MTs growing toward the cell poles; 2) stabilization of astral MTs aimed toward the cell equator and delivering centralspindlin to the equatorial cortex; 3) stalling of centralspindlin upon reaching MT plus ends. Though accumulating data cast doubt onto these assumptions, the value of this and previous models are in positing precise hypotheses for the process of experimental discovery. One glaring omission in all existing models of the furrow positioning is that these models do not consider that overlapping, antiparallel MTs create special binding sites that might be relevant to localizing regulatory proteins both in the midzone and the equatorial cortex between the spindle poles.

### 5.2. How Does the Contractile Ring Assemble?

Modeling helps to understand how a contractile structure at the cell equator self-assembles from a disordered array of actin and myosin filaments. Theoretically, this process is best understood for contractile ring in fission yeast. A very detailed, geometrically realistic search-and-capture model, which was proposed in [120] and refined in [121], posits that actin filaments grow with their pointed ends outward from formin–myosin nodes, get captured by other nodes and coalesce into the ring-like contractile structure under a wide range of mechanical conditions. There are other, competing models, which did not attract the same level of computational analysis. In animal cells, the process is likely more complex, and therefore less clear. It was shown recently that initial ring assembly and contraction causes flow of the cortex into the ring, which leads to actin filament alignment and enhanced myosin-powered contraction, reinforcing the ring development [122]. One recent theory [123] made the first inroad into modeling the initial stage of the contractile network self-assembly by simulating flexible actin and myosin filaments and crosslinkers and demonstrating gradual coalescence of the initially disordered network into an anisotropic contractile structures.

### 5.3. How Does the Ring Generate Force?

The fundamental puzzle is how the disordered array of actin and myosin filaments and crosslinkers can contract? One earlier theoretical proposal is a myosin-independent Brownian ratchet mechanism with actin filament pointed-end disassembly coupled with flexible cross-link expanding, “grabbing” another filament and pulling two filaments together thereby producing a contractile force [124]. Most models, however, assume myosin-powered sliding of actin filaments, which goes back to the original discoveries of actin filaments [125] and myosin-II [2] in the cleavage furrow and the demonstration that myosin antibodies stop constriction [53]. A molecularly explicit model of the fission yeast contractile ring [126] is based on the assumption, well supported by data, that formin nodes generate actin filament asters with pointed ends outward. When filaments from neighboring asters overlap, myosin clusters act on the antiparallel actin filaments creating ‘disheveled’ mini-sarcomeres and causing contraction. There are a number of competing models summarized in the introduction to a recent theoretical paper [127]. In this paper, the authors used comprehensive computational screening to reveal that multiple mechanisms, including sliding apart and buckling overlapping barbed ends and contracting overlapping pointed ends, reorganizing the disordered actin-myosin arrays into asymmetric contractile networks. Experiments have yet to sort out what combination of the proposed theoretical mechanisms work in which cell types.

### 5.4. How Does this Force Form the Furrow?

The greatest number of modeling projects on cytokinesis were devoted to the question that fascinated experimentalists and theoreticians equally: how should the inhomogeneous and anisotropic stresses in the cell cortex be distributed to generate the characteristic dumbbell cell shape? A number of models reconstructed stress distributions necessary to generate the observed shapes of dividing cells, assuming various viscoelastic mechanical properties of the cell [128,129,130,131,132,133].

Are there essential feedbacks between the cell geometry, cortex dynamics, and actomyosin network architecture? If there is an initial uniform distribution of contractile elements around the cell cortex, and then the contractility is relaxed at the poles, would the model predict the observed dumbbells shape [116]? The White–Borisy model gave a negative answer because the saddle-like curvature of the furrow at the cell equator weakens the effective force that deepens the furrow. White and Borisy then realized that full cell division can be explained if contractile elements are pulled into the cell equator by actin flow and align along the equator. A number of recent, much more sophisticated, computational models extended this line of thinking and showed that positive feedbacks between the cytoskeletal alignment, flow of the cortex into the furrow, cell surface curvature, and actomyosin contractility are not only sufficient, but also necessary to explain the spatiotemporal progression of cytokinesis [122,134,135,136]. A few of these studies, in fact, combined modeling and experiment [122,128,129,132,136] to test the predictions and to refine the models. Very sophisticated numerical methods of applied mathematics have also been used to mimic the 3D shapes of the dividing cells [137,138,139].

Some important questions—for example, about expansion of the membrane during the furrow ingression, about membrane-actin interactions in the contractile ring, about daughter cells separation, etc.—have not attracted theoreticians’ attention yet and await future modeling efforts. There has also been little modeling effort to elucidate mechanisms of biochemical regulation of the cortex in cytokinesis. One exception is the recent detailed model of the actin cortex that included signaling of Rho GTPases governing the cortical mechanics and actin dynamics, such that Rho was an activator and actin was an inhibitor [140]. The model predicted a highly nontrivial oscillatory, excitable behavior at the onset of cytokinesis, which was confirmed experimentally. This study creates a quandary: how can steadily contractile furrow co-exist with the excitable waves in the rest of the cortex? Goryachev et al. speculated that within the strip of the equatorial cortex, where the ECT2 concentration is above the critical, the wiring of the cortical network of biochemical reactions is tuned by elevated ECT2 (or by other signaling molecules, such as Aurora B kinase and Plk1, confined to the narrow equatorial region), in such a way that only one steady-state is possible, with high Rho activity and actin accumulation [141]. The most important future challenge for modeling is to decipher feedbacks between the cortex signaling dynamics, contractile mechanics, and spindle regulation that underlie robust cytokinesis.

## 6. Midzone-Independent Signaling in Cleavage Plane Specification

Since so many cytokinetic regulators localize to the midzone, the field has long focused heavily on midzones as the key determinant of positing the cleavage furrow. However, in recent years, it has become evident that midzone-independent signaling also plays a crucial role in cleavage furrow assembly and cytokinesis. In *Drosophila S2* cells, we recently showed that many key regulators of cytokinesis (centralspindlin complex, ABK, and Polo kinase) localize to and track MT plus-ends [62]. Interestingly, ECT2 was recruited to cortical contact sites within seconds of contact by astral MT plus-ends. Localized RhoA activation followed seconds later. RhoA activation further resulted in accumulation of myosin-II at the equatorial cortex and furrow ingression within minutes [62]. Localization of centralspindlin complex to the MT plus-tips depended on a putative EB1-interaction motif (hxxPTxh) which is located at the C-terminus of *Drosophila* MgcRacGAP/Tumbleweed. Deletion of this novel EB1-interaction motif (hxxPTxh) in the Tumbleweed resulted in higher incidences of cytokinesis failure, leading to the proposal that EB1-dependent, MT plus-tip-based signaling hubs recruit cortical ECT2 upon contact to locally activate RhoA [62].

Localization of centralspindlin complex to the MT plus-tips has also been observed in other organisms [62,88,142,143] indicating that this mode of signaling for cleavage furrow initiation is conserved between flies and vertebrates. Why is there a need to employ a signaling mechanism that is independent of the midzone and primarily dependent on astral microtubule plus-ends? It is conceivable that cells operate parallel mechanisms of signaling to establish and maintain a robust cleavage furrow. The advantage of MT plus-end-based signaling is that it does not have to rely on diffusion, and signals can be directly delivered to the cortex through physical contact, which is more reliable and robust; especially in large cells where signals may need to be delivered over hundreds of microns.

Further evidence for spindle midzone-independent signaling was provided by Kotykova et al. In contrast to the current notion, which presumes ECT2 recruitment to the midzone is essential for cleavage plane specification, this study showed that artificial targeting of ECT2 to the plasma membrane was sufficient for cytokinesis [99]—i.e., ECT2 recruitment to the spindle midzone can be dispensable, but its localization to the plasma membrane is necessary for cleavage furrow induction. In *C. elegans* embryos a NOP1-mediated pathway can activate ECT2 in the absence of centralspindlin complex [144]. However, the mechanism through which Nop1 activates ECT2 remains elusive [94].

A key regulator of cytokinesis is PRC1. It localizes to the nucleus during interphase; however, during the metaphase-to-anaphase transition it localizes to the spindle midzone through its binding partner KIF4A [145]. Given its critical role in cell cycle regulation, the spatiotemporal localization of PRC1 is tightly controlled by phospho-regulation [145,146,147]. At the spindle midzone PRC1 binds to the centralspindlin complex and bundles anti-parallel microtubules to strengthen the assembly of spindle midzone organization [50,148,149]. PRC1 and centralspindlin complex at the midzone recruit key regulatory proteins to initiate the formation of the cleavage furrow. RNAi mediated depletion or knockdown of PRC1 has been shown to disrupt the spindle midzone, but it does not impede RhoA activation and cleavage furrow ingression during cytokinesis [62,146,150], leading to the proposition that spindle midzone-independent signaling can locally activate RhoA and induce cleavage furrow ingression. In a recent study, Adriaans et al., characterized a spindle midzone independent pathway for cleavage furrow ingression by knocking down PRC1. The pathway was found to depend upon ABK activity and centralspindlin at the equatorial cortex; however, this midzone-independent pathway could function in the absence of PLK1 activity [151]. Thus, with regards to midzone-independent mechanisms, polo kinase activity may be dispensable for the cortical ABK- and centralspindlin-based mechanism described by Adriaans et al., but is essential for the MT plus-end based RhoA activation mechanism described in the next section [62].

## 7. Astral Stimulation and Polar Relaxation: Two Sides of the Same Coin?

We recently characterized an astral stimulation mechanism in *Drosophila* S2 cells that is mediated by key cytokinesis regulators that localize to and track the growing plus-ends of astral MTs within minutes of anaphase onset [62]. We deemed these specialized MT plus-ends cytokinesis signaling (CS)-TIPs because they recruited ECT2 to cortical sites within seconds of physically contacting the plasma membrane. ECT2 recruitment to the contact sites, in turn, resulted in rapid (seconds time-scale) activation of RhoA in the immediate vicinity of CS-TIPs. Sustained signaling by CS-TIPs over a minutes-long time-scale led to localized enrichment of myosin and cortical contractility. CS-TIP assembly required global ABK activity while recruitment of ECT2 and RhoA activation by CS-TIPs required global polo kinase activity. Neither CS-TIP assembly nor signaling required midzone-localized ABK or midzone-localized polo kinase. We proposed that the CS-TIP-based signaling pathway is mediated by recruitment of membrane-associated ECT2 by the MT plus-end associated centralspindlin complex, specifically via direct binding of ECT2 to MgcRacGAP, the affinity of which is increased by polo-dependent phosphorylation of MgcRacGAP [60,61,62,96,110].

During early anaphase, CS-TIPs assembled uniformly on astral MTs—both in the polar and equatorial regions of the cell (Figure 2A). However, after ~10 minutes, when the cell was in late anaphase/telophase, CS-TIPs, which were retained on equatorial astral MTs, disassembled from polar astral MTs—an unexplained phenomenon that we called CS-TIP patterning (Figure 2B). On its face the CS-TIP-based signaling mechanism is an astral stimulation model; however, the patterning phenomenon in which polar CS-TIPs disassemble is also inherently a polar relaxation mechanism because it suppresses the ability of polar astral MTs to signal and promote cortical contractility. The mechanism of CS-TIP patterning is presently unknown, but there are several appealing models that we will outline here that evoke spatial gradients (Figure 2).

Chromosome or kinetochore-based gradients could play a role by locally inhibiting CS-TIP assembly as the segregating chromatids approach the polar cortex and envelop the polar astral MTs. The RanGTP gradient [152] has been shown to negatively regulate the cortical localization of spindle positioning molecules in the vicinity of chromosomes [153]. Another gradient that has been shown to contribute to polar relaxation is a PP1-Sds22 phosphatase activity gradient that is proposed to emanate from kinetochores and dephosphorylate the membrane stiffening molecule moesin to locally relax the polar cortex during anaphase [46,47]. It is possible that the RanGTP gradient and/or PP1 activity gradients locally inhibit CS-TIP assembly as sister chromatids move poleward during anaphase albeit priority should be given to investigating the phosphatase activity gradient since we know that ABK activity is required for CS-TIP assembly and it is well-established that PP1 opposes ABK [154].

Even if chromosome- or kinetochore-derived gradients promote polar CS-TIP assembly, other mechanisms must also contribute since CS-TIP patterning can occur during telophase after the nuclear envelope reforms and kinetochores disassemble. Pole/centrosome-based activity gradients are another possible source of the polar CS-TIP disassembly signal (Figure 2). We previously visualized a polar AAK activity gradient that extends microns from the centrosome/spindle pole in *Drosophila* S2 cells [155]. Prior to anaphase the polar AAK activity gradient regulates kinetochore-MT attachments [80,81], but it also functions to spatially regulate important cellular phenomena after anaphase onset. In fact, a recent study conducted in *C. elegans* proposed that a diffusible AAK activity gradient generated by the AAK-activating MAP TPXL-1 (*C. elegans* TPX2) contributes to polar relaxation by clearing contractile ring proteins from the polar cortex [45]. While it is possible that an AAK activity gradient contributes to polar relaxation by removing contractile ring components in *Drosophila* cells, we feel that it is unlikely to promote polar CS-TIP disassembly through the same mechanism since 1) the *Drosophila* TPX2 homolog does not contain the AAK activating domain [156]; and 2) ABK, which shares many substrates with AAK, promotes CS-TIP assembly not disassembly. Polo kinase has also been proposed to produce a polar activity gradient that negatively regulates the localization of cortical dynein [153], but it too is unlikely to contribute to disassembly of CS-TIPs since it promotes their ability to recruit ECT2 and activate RhoA [62]. Thus, it will be worthwhile to further investigate if polar components capable of generating activity gradients contribute to polar relaxation via disassembly of polar CS-TIPs during late anaphase and telophase.

## 8. Propagating Waves of Contractility in Cytokinesis

In animal cells, RhoA plays a vital role in establishment of the actomyosin contractile ring via its downstream effectors formins and Rho kinases (Figure 1E). Formins nucleate and elongate actin filaments and Rho kinases phosphorylate myosin-II regulatory light chains, which activates the assembly of myosin-II as well as the actin-myosin-II ATPase cycle [157,158]. Centralspindlin along with other regulatory proteins—such as, ABK, PLK1, ECT2, and RhoA—pattern the formation of the cleavage furrow, and it is now widely accepted that in animal cells the whole cortex can be responsive to the signaling cues emanating from spindle MTs. Recent evidence indicates that cortical excitability is often mediated by travelling waves of RhoA and actin filaments [140,141,159]. However, our understanding of the nature and origin of travelling waves during cytokinesis is far from complete, and the means by which proposed RhoA activity waves are translated into a cleavage furrow at a specific location during cytokinesis is unclear. Multiple reports evoke an activator–inhibitor mechanism originally proposed by mathematician Alan Turing to explain the propagation of RhoA activity waves. [140,159]. More specifically, the model posits that waves are a consequence of a combination of 1) an autocatalytic positive feedback, exhibited by RhoA, at the wave front and 2) a negative feedback mechanism that inhibits RhoA at the back of the wave, which is mediated by an unknown actin filament-associated RhoA inhibitor or GAP. Bement et al. proposed that a reaction-diffusion system can explain the co-existence of both wave-like and static behaviors of Rho and actin during cytokinesis. The assumption of a reaction-diffusion system is that the activator (Rho) auto-amplifies through a positive feedback mechanism and generates its own inhibitor (actin), which then impedes the propagation of activator. In the case of travelling waves—the activator, RhoA, always precedes the inhibitor, actin, and in the case of static profile as seen in cleavage furrow, the activator (RhoA) and the inhibitor (actin) accumulate at the same rate [141]. The proposed hypothesis was experimentally tested by injecting C3 exotransferase (a bacterial toxin that inactivates RhoA), which abolished actin waves. Additionally, expression of a dominant negative mutant of ECT2 also resulted in loss of actin waves, leading to the conclusion that RhoA was upstream of the actin waves [140]. Two-color imaging of Rho and actin filaments revealed that actin followed Rho activity waves in starfish zygotes and frog blastomeres [140]. Furthermore, two drug treatments directly tested the idea that accumulation of actin filaments negatively regulates Rho; stabilizing actin filaments with jasplakinolide diminished Rho waves, while depolymerizing actin with latrunculin B amplified Rho activity [140]. Furthermore, Rho-waves were lost in frog and starfish eggs microinjected with mRNA encoding ∆90 cyclin B to prevent Cdk1 proteolysis [160], leading [140] to conclude that Cdk1 regulates Rho-actin waves.

Wave-like dynamics of Rho activity was also recently reported in human U2OS osteosarcoma cells [159]. Travelling waves of Rho activity were amplified via the recruitment of its activator GEF-H1 and the activity of Rho was suppressed by the action of Myosin-IIa and a RhoGAP Myo9b, which act at different time-points [159]. In contrast to *X. laevis* and *C. elegans* data where ECT2 activity is required for excitable dynamics of RhoA [140,161], this study concluded that GEF-H1, which belongs to the Lbc family of GEFs, is required for propagation of Rho waves. Travelling waves of Rho activity were also observed in primary human endothelial cells [162] and *C. elegans* zygote [161], indicating that propagating waves of Rho activity is a conserved feature in different cell types and organisms. Altogether, these experiments support the idea that RhoA, ECT2, and actin are critical components of travelling waves, but they do not explain: 1) self-activation of activator (RhoA); and 2) inhibition of RhoA by actin.

An alternative and more intuitive model for propagating RhoA activity waves is called- local-excitation, global-inhibition (LEGI) [163,164]. In the LEGI model, actin-propelled RhoA activity waves would be generated if 1) RhoA is distributed uniformly throughout the plasma membrane, 2) Rho GAPs (global inhibitors) are distributed uniformly throughout the cytoplasm and/or plasma membrane, and 3) membrane-associated RhoGEFs such as ECT2 (localized activators) are propelled within the inner leaflet of the plasma membrane by actin polymerization. In this case, RhoA would be activated by the localized activator (ECT2) at the wave-front, and would then become inactivated after some period of time by the global Rho GAPs after the wave-front had passed. Interestingly, Graessl et al. observed that the Rho activator GEF-H1 itself exhibited wave-like propagation at the plasma membrane [159]. While the authors argued that Rho activity waves required a positive feedback loop at the wave-front that was mediated by binding of GEF-H1 to active RhoA, the GEF-H1 peak intensity actually preceded the Rho activation peak at the wave-front by 2.5 seconds. This observation is more supportive of a LEGI model than an activator-inhibitor mechanism. It would be worthwhile to determine if, like GEF-H1, ECT2 also exhibits wave-like propagation at the plasma membrane during cell division.

## 9. Architecture of the Contractile Ring, Ring Constriction, and Furrow Formation

The inventory of proteins that form the contractile ring is well known; however, understanding of the nanoscale architecture of the contractile ring is far from complete. So far, the best-known system where the architecture of contractile ring has been elucidated is *S. pombe* [165,166]. [165] reported that in the contractile ring, proteins were distributed into three layers: the proximal layer (0–80 nm from the membrane) contained formin, tail of the myosin-II, and membrane binding proteins that anchor the ring. The intermediate layer (80–160 nm) was composed of accessory and signaling proteins, and the last layer (160–350 nm) was composed of actin filaments and the motor domains of myosin. Further insight into 3D organization of individual actin filaments within the contractile ring came from an electron cryotomography study [167]. Authors reported that actin filaments ran parallel to each other and to the membrane, but did not associate with the membrane directly. The actin bundles were composed of 35 filaments on average, and the average distance between neighboring actin filaments was 11.5 nm. However, the authors were not able to resolve or detect the distribution of any regulatory proteins or myosin-II in this study. In recent years, the nanoscale organization of the contractile ring in animal cells has also improved [168,169,170]. Henson et al. combined super-resolution light microscopy and transmission electron microscopy to visualize the ultrastructure of actin filaments and myosin-II in sea urchin embryos. They reported that actin filaments and myosin-II were tightly packed in a linear array within the contractile ring [169], however the resolution was not enough to differentiate between individual actin filaments and myosin-II. The authors argued that organization of actin and myosin-II is consistent with the sliding filament-based mechanism of ring constriction. Whether a similar mechanism of ring constriction operates in other organisms and cell types remains to be established.

In order to form the cleavage furrow, the contractile ring must be physically linked to the plasma membrane. We do not have a complete inventory of proteins that physically establish this connection. Anillin is a conserved multi-domain protein that contains three membrane-binding domains [171], and it has been shown to bind to MgcRacGAP [172], ECT2 [173], RhoA [174], myosin-II [175], and actin filaments [176]. Thus, it appears to act as a scaffolding protein that bridges the central spindle and actomyosin ring to the plasma membrane. In addition to anillin, F-BAR proteins are known to link membranes to the actin filament. In a recent study, *S. pombe* Cdc15, which contains F-BAR domain, was shown to bind membranes both in vivo and in vitro, and oligomerization was shown to be critical for membrane binding [177]. Mutations in the F-BAR domain that compromised membrane binding also resulted in cytokinesis failure [177]. Interestingly, six human F-BAR proteins were identified in this study which showed membrane binding activities; however, their mechanism of action in cytokinesis and how they tether the contractile ring to the plasma membrane remain unresolved.

## 10. Summary and Outlook

Cytokinesis in animal cells can be categorized into three major steps: 1) cleavage plane specification, 2) formation and maintenance of the actomyosin ring, 3) constriction of the ring to produce two daughter cells. Although multiple mechanisms of cleavage plane positioning have been proposed, central to all these mechanisms are MTs. Cells clearly employ redundant MT-dependent signaling pathways to position the cleavage furrow. How disparate signaling mechanisms coordinate to define the cleavage plane will be an interesting area of investigation. We also do not fully understand how the actomyosin ring generates the contractile force required to make the cleavage furrow nor how it remains physically linked to the membrane as the furrow ingresses. We also have a limited understanding of the architecture of the contractile ring. As we gain information about force-generation and the architecture of the contractile ring, it will be important to integrate mathematical modeling to build more comprehensive and accurate models of these fundamental cellular processes. Additionally, biochemical reconstitution of cytokinesis using purified proteins and in vitro systems will provide mechanistic insights into the steps through which cleavage plane specification, force-generation, and ring constriction occur.

## Figures and Tables

**Figure 1 biology-08-00055-f001:**
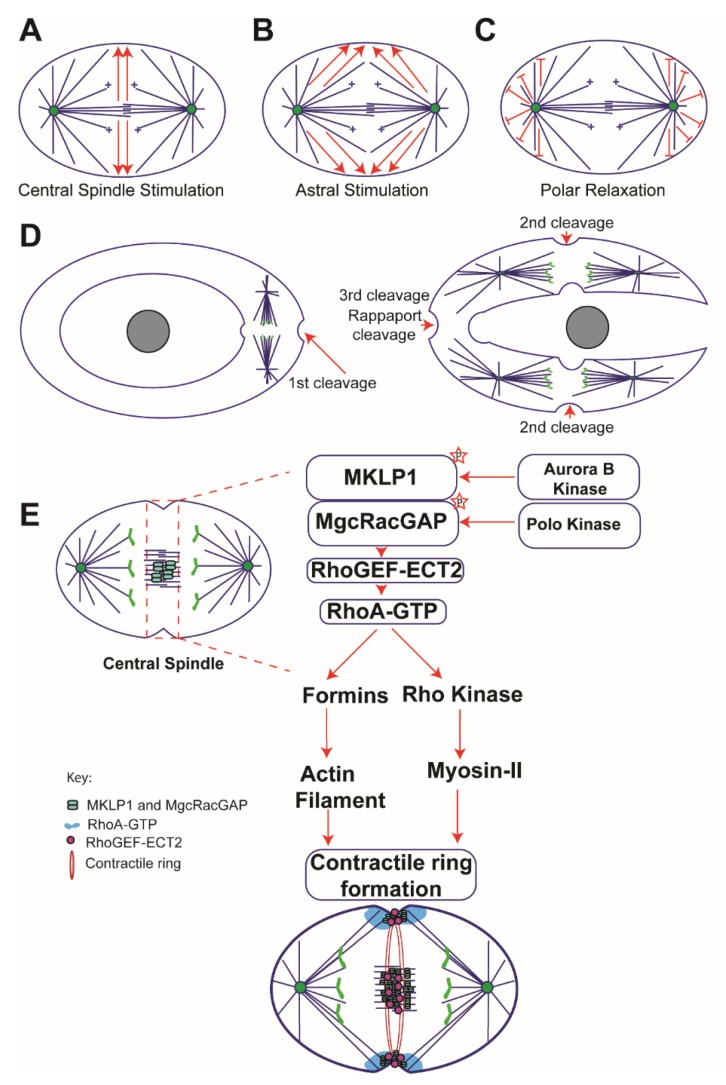
Classical models of cytokinesis: Schematic representation of three different models that have been proposed to position the cleavage furrow. (**A**) Central spindle stimulation model: according to this model a diffusive signal emanating from the central spindle/midzone activates RhoA, which then initiates actomyosin ring formation. (**B**) Astral stimulation model: this model posits that astral microtubules provide positive signaling cues for furrow initiation. (**C**) Polar relaxation model: this model presumes that MTs near the poles and/or segregating chromosomes carry the inhibitory signals that restrict cortical contractility in the nearby region. (**D**) Illustration of the famous Rappaport “torus experiment” in sand dollar eggs, arrow heads indicate cleavage sites and gray circle in the middle represents perforation site. (**E**) A model depicting the composition of the central spindle and assembly of the contractile ring: the centralspindlin complex, which is composed of MKLP1 and MgcRacGAP, localizes to the central spindle during anaphase. Phosphorylation of MgcRacGAP by PLK 1 generates a docking site for the RhoGEF, ECT2, at the central spindle. ECT2 activates RhoA at the equatorial cortex, which then triggers a downstream signaling cascade via Formins and Rho kinase resulting in the assembly of an actomyosin ring. Constriction of the ring results in generation of two daughter cells.

**Figure 2 biology-08-00055-f002:**
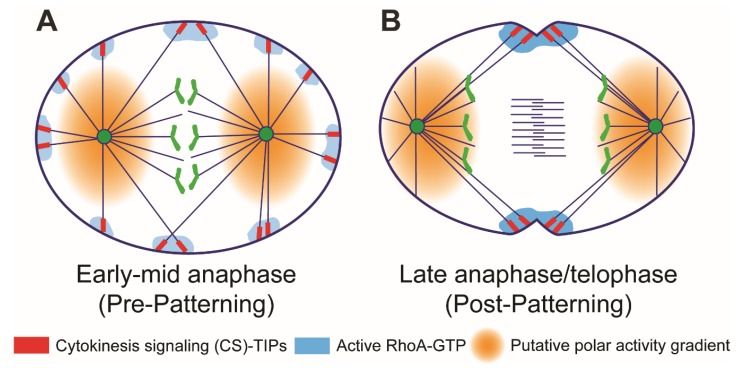
Two sides of the same coin? (**A**) Cytokinesis signaling molecules localize to the plus-ends of astral MTs (referred to as cytokinesis signaling (CS)-TIPs; shown in red) within minutes of anaphase onset and activate RhoA (shown in blue) upon cortical contact. In this model, a putative polar gradient (shown in orange gradient) inhibits the localization of CS-TIP components to MT plus-ends. During the “Pre-Patterning” phase in early-mid anaphase, the polar gradient does not reach astral MT plus-ends and; therefore, astral MT plus-ends in both the polar and equatorial regions are capable of triggering transient RhoA activation. (**B**) The unexplained phenomenon of CS-TIP patterning involves the retention of CS-TIPs on equatorial astral MTs and the loss of CS-TIPs from polar astral MTs during late anaphase and telophase (referred to as ‘Post-Patterning’). In this model, polar CS-TIPs are lost as the spindle poles approach the polar cortex and the inhibitory polar gradient envelops polar astral MT plus-ends. Since the inhibitory polar gradient still does not reach the equatorial MT plus-ends, CS-TIPs in this area are retained and sustained RhoA activation supports cleavage furrow assembly and ingression in the equatorial region.

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
