# Peer review of "Classical and Emerging Regulatory Mechanisms of Cytokinesis in Animal Cells"

_biology, 2019, doi:10.3390/biology8030055_

Round 1
Reviewer 1 Report
Verma, Mogilner and Maresca
Cytokinesis in animal cells – how cells decide where to establish the cleavage plane. This review is about much more, so the title is misleading.
Overall, this is an interesting review of what we know and do not know about cytokinesis. It has the potential to be even better, if the authors are willing to make a through revision.
Content: This an account of the history of experiments on cytokinesis, which is both a strength and a weakness. On the positive side, it reminded me of many historically interesting experiments. On the other hand, much of the text simply repeats what various investigators wrote about their data and interpretation, without much evaluation of the strengths and weaknesses of that work. One exception is lines 644-658, one of the best parts of the paper. Here the authors make a thoughtful argument about ideas in the field rather than just repeating what other have written. More of such evaluative material like this throughout would make this review much more valuable.
Style: The paper is full of awkward constructions and grammatical errors (many but not all pointed out below). The authors make heavy use of nouns as adjectives, for example “Centralspindlin plus-tip localization.” Consider fixing throughout.
Much of the text is wordy. Here is one example starting on line 622: “To directly test the idea that accumulation of F-actin negatively regulates Rho; cells were treated with jasplakinolide which stabilizes F-actin. This treatment resulted in diminished Rho waves. Further, cells treated with latrunculin B which inhibits polymerization of F-actin resulted in amplification of Rho activity (Bement et al., 2015). Following these experimental outcomes, experiments were further set out to see if Rho-actin waves were dependent on cell cycle stages. Cells derived from frogs and starfish were microinjected with mRNA encoding ∆90 cyclin B, which has been shown to stop Cdk1 inactivation (Murray and Kirschner, 1989), resulted in loss of Rho-actin waves (Bement et al., 2015); leading to the conclusion that Cdk1 regulates Rho-actin waves.” (113 words)
Here is a more concise version: “Two drug treatments directly tested the idea that accumulation of actin filaments negatively regulates Rho; stabilizing actin filaments with jasplakinolide diminished Rho waves, while depolymerizing actin with latrunculin B amplified Rho activity (Bement et al., 2015). Furthermore, Rho-waves were lost in frog and starfish eggs microinjected with mRNA encoding ∆90 cyclin B to prevent Cdk1 proteolysis (Murray and Kirschner, 1989), leading Bement et al. (2015) to conclude that Cdk1 regulates Rho-actin waves.” (71 words)
The authors should consider using similar strategies throughout to free up space to evaluate the work being reviewed.
Specific comments and suggestions:
“Actin filament” is preferred to “F-actin”, which is jargon.
Line 11: Think about it; the contractile ring is really not a “donut-shaped structure.”
Line 25: cytoplasm is preferred to cytosol.
Line 30: I would say “major feature of cancer development.”
Line 41: rewrite “and how these signals are delivered.”
Line 57: rewrite “…middle of the cell). Note that the term “central spindle’’ and …..
Lines 80-98: This paragraph has grammatical errors and awkwardly constructed sentences.
Line 155: Very awkward: “Although, multiple evidences have been provided in support and against of every model.” The rest of the paragraph is the key point.
Line 158: Fig 1D. My understanding of this signaling pathway (see Pollard, JCB, 2017, Fig 3) differs from this diagram. The RhoA cycle with an arrow from Ect2 will confuse some readers. I would delete this cycle or incorporate it to the main pathway and change the labels from “Ect2” to “Ect2-GEF” and “RhoA” to “RhoA-GTP”. Note also that actin filaments are always “linear.” The point is that those polymerized by a formin are not branched. The floating, disconnected microtubules in the middle of the bottom cell are unrealistic and possibly confusing.
Line 167: The following may be confusing: “a kinesin like protein MKLP1 and a GTPase activating protein MgcRacGAP binds to the central spindle, which then recruits a RhoGEF ECT2 and other signaling molecules. ECT2 then activates RhoA.” The key event is ECT2 localized around the equator activating RhoA locally (as explained later on line 274). How ECT2 gets from the central spindle to its site of action in the equatorial cortex is still being investigated, as explained later.
Lines 187-194: Although this is in part our work, more research is needed to confirm that Cdks really inhibit myosin-II during mitosis.
Line 212: grammar problems “factors that recruit Plk1 to the central spindle was not known,” “This indicates that factor that recruit Polo kinase to the MT plus-ends is not mediated by PRC1 in Drosophila.” “that were defective in centrosome cycle…”
Line 223: Break up the long paragraph into smaller parts. This is a thorough account of the history, but it needs clearer conclusions, so the reader comes away knowing the state of the art.
Line 279: rewrite “…caused cytokinesis to fail owing to defects in contractile ring assembly ...”
Line 280: awkward “Phenotypes displayed by ECT2 depletions resembled RhoA inactivation, which led to the proposal it acts upstream of RhoA.” What is “RhoA inactivation?” How about “Depleting ECT2 or inactivating RhoA cause similar phenotypes, suggesting that ECT2 acts upstream of RhoA?” But why not the other order?
Line 283: rewrite “One is that ECT2 is subject to auto-inhibition.”
Line 288: What is a “Phospho-null mutant?”
Line 289: unclear and possibly misleading “Phospho-null mutants of MgcRacGAP prevented stable assembly of the centralspindlin complex and ECT2…” Does this mean that the mutants prevented the assembly of the heterotetrameric centralspindlin complex or the binding of ECT2 to the centralspindlin complex?
Line 290: I think the following is wrong: “loss of ECT2 from midzone and inactivation of RhoA.” The loss of ECT2 from midzone does not inactivate RhoA; rather it results in the failure of RhoA to be activated, which is different. Line 317 has the same problem.
Line 299: What is the nature of the disagreement in “Disagreement initially originated…”?
Line 300: awkward “MgcRacGAP has stronger GAP activity towards the GTPases CDC42 and RAC1 than RhoA.”
Line 295-353: As it stands, all of these apparently contradictory studies are given equal weight. The authors really need to weigh in on the quality of this work, since some studies are much better than others. To cite one example among many, did a given paper show that certain mutations changed the enzyme activity of MagRacGAP or might they have caused folding defects that would influence not only enzyme activity but also interactions with ECT2? My conclusion from all of these studies is that both the RacGAP and ECT2 activation functions contribute in different ways to cytokinesis. See Pollard, JCB, 2017, Fig 3.
Line 382: Sentence needs a subject.
Line 386: rewrite “This modified model predicted global signal maxima at the equator…”
Line 392: Why is this model the state of the art?
Line 399: None of these models seem to consider that overlapping, antiparallel microtubules create special binding sites that might be relevant to localizing regulatory proteins both in the midzone and the equatorial cortex between the spindle poles. This only comes up at line 500, where binding to antiparallel microtubules is only mentioned in passing and the focus is on disruption of the central spindle. Does not the loss of PRC1 impact the antiparallel microtubules coming from the poles?
Line 407: One really must cite Vavylonis Science (2008), which established most of the important ideas that were later refined in the Bidone paper. Vavylonis 2008 was the “first inroad.” One should also mention the work of Reymann et al. eLife (2016), on contractile ring formation in animal cells.
Line 411: One might note that sliding filament models go all the way back to the original discoveries of actin filaments (Schroeder) and myosin-II (Fujiwara) in the cleavage furrow and the demonstration that myosin antibodies stop constriction (Mabuchi).
Line 416: The Stachowiak paper is founded on much more than an “idea”, since it was motivated by extensive data on the numbers, biochemical activities and localizations of the component molecules and tested by observations of the motions of the molecules and effects of mutations on ring constriction.
Line 433-466: grammar problems. “Important relevant question is:” “One of the earliest relevant investigation, by..”, “what are the mechanical requirements to distribution of inhomogeneous and anisotropic stresses…” and “to help the experiment to decipher feedbacks.”
Line 475: it is confusing to say “accumulation of myosin-regulatory light chain (MRLC)” because this is merely an marker for myosin-II, which is the relevant molecule.
Line 478-479: Was the protein stable after deleting the hxxPTxh motif? That is a crucial control.
Line 498: grammar problem “mechanism through which Nop1 activate ECT2 remains elusive.”
Line 596: misleading “Formins help to facilitate F-actin polymerization and Rho kinases activate the assembly of myosin-II.” Formins nucleate and elongate actin filaments not “help to facilitate F-actin polymerization” and Rho kinases phosphorylate myosin-II regulatory light chains which not only “activates the assembly of myosin-II” but also activates the actin-myosin-II ATPase cycle.
Line 597: grammar problem and misleading “proteins belonging to small GTPase family.” It is not the whole small GTPase family, which has diverse members participating in many other aspects of cellular function.
Line 608: rewrite “inhibits RhoA at the back of the wave” not “end.”
Line 634: myosin IIa is not a proper noun. What is the explanation of “In contrast to the notion that RhoGEF Ect2 activity is required for excitable dynamics of RhoA?” Different cell types? What about “primary human endothelial cells and C elegans zygotes?”
Line 649: It is incorrect to say “are propelled THROUGH the plasma membrane by F-actin polymerization.”
Line 668: Only at the very end do the authors consider the crucial point “how it remains physically linked to the membrane.” More on this and its importance would strengthen the contraction section.
Reviewer 2 Report
The regulation of cytokinesis in animal cells is a very complex and nuanced process and this review does a reasonable job of summarizing the current state of understanding in the field. In addition to addressing material covered in other recent reviews, this paper’s provides important context for the interesting recent findings of the authors concerning MT plus ends acting as signaling hubs for Rho activation during cytokinesis. There is also an attempt to review the work on computational modeling in cytokinesis as well as the studies demonstrating the potential role for propagating waves of contractility in cytokinesis. Below are some specific comments / suggestions:
- One important recent review article that needs to be referenced in this paper concerns the regulation of RhoA in cytokinesis (Basant and Glotzer, 2018. Curr Biol28: R570-R580).
- The seminal work of Rappaport is discussed, however it is described as being accomplished in “sea urchin eggs” (line 36) when the vast majority of his studies were performed on sand dollar embryos. In addition, the explanation provided for Rappaport’s famous torus experiment (lines 118-130) is not clear and should be rewritten.
- The authors indicate that Cdk1 activity inhibits myosin II ATPase activity until after chromosome segregation (lines 188-190), however Shuster and Burgess (2002. Curr Biol14: 854-858) demonstrated that this was not the case, at least in echinoderm embryos.
- Some parts of the paper suffer from less than clear writing style and sentence structure. The section on computational modeling (starting on line 365) needs particular attention. For example, the first sentence needs to indicate that Pollard’s nine unanswered questions were related to cytokinesis.
- On occasion this review refers to the structure of the contractile ring, including a statement in the summary referencing a need to gather new information about the architecture of the contractile ring (lines 669-671). It should be noted that there have been three recent papers that have improved our understanding of the architecture of the contractile ring in both living and fixed cells. These are from Beach et al. (2014. Curr Biol 24: 1160-1166), Fenix et al. (2016. Mol Biol Cell27: 1465-1478), and Henson et al. (2017. Mol Biol Cell28: 613-623).
Reviewer 3 Report
In the manuscript entitled "Cytokinesis in animal cells – how cells decide where to establish the cleavage plane", Verma et al. summarized key findings in understanding of the molecular mechanisms that, on the one hand, control the formation of the cleavage furrow and, on the other, cytokinesis.
The manuscript is interesting in the field and states a good source for further citations. I suggest to publish it in Biology without any further modifications.
Author Response
Reviewer 3
In the manuscript entitled "Cytokinesis in animal cells – how cells decide where to establish the cleavage plane", Verma et al. summarized key findings in understanding of the molecular mechanisms that, on the one hand, control the formation of the cleavage furrow and, on the other, cytokinesis.
The manuscript is interesting in the field and states a good source for further citations. I suggest to publish it in Biology without any further modifications.
Response: We would like to thank the reviewer for his/her positive comment on the manuscript.
Round 2
Reviewer 1 Report
The authors responded constructively to my concerns about the original manuscript, so this interesting, scholarly review is now appropriate for publication. A few minor problems remain in the text, so another round of careful proofreading is recommended. For example, the Abstract should read "Decades of research HAVE identified..." On page 17 "anillin" is not a proper noun and should not be capitalized.
Tom Pollard